# A Systematic Review and Meta-Analysis of Immunoglobulin G Abnormalities and the Therapeutic Use of Intravenous Immunoglobulins (IVIG) in Autism Spectrum Disorder

**DOI:** 10.3390/jpm11060488

**Published:** 2021-05-30

**Authors:** Daniel A Rossignol, Richard E Frye

**Affiliations:** 1Rossignol Medical Center, 24541 Pacific Park Drive, Suite 210, Aliso Viejo, CA 92656, USA; 2Barrow Neurological Institute at Phoenix Children’s Hospital, 1919 E Thomas Rd, Phoenix, AZ 85016, USA; rfrye@phoenixchildrens.com

**Keywords:** autism spectrum disorder, immunoglobulin G, intravenous immunoglobulin

## Abstract

Autism spectrum disorder (ASD) is a neurodevelopmental disorder affecting approximately 2% of children in the United States. Growing evidence suggests that immune dysregulation is associated with ASD. One immunomodulatory treatment that has been studied in ASD is intravenous immunoglobulins (IVIG). This systematic review and meta-analysis examined the studies which assessed immunoglobulin G (IgG) concentrations and the therapeutic use of IVIG for individuals with ASD. Twelve studies that examined IgG levels suggested abnormalities in total IgG and IgG 4 subclass concentrations, with concentrations in these IgGs related to aberrant behavior and social impairments, respectively. Meta-analysis supported possible subsets of children with ASD with low total IgG and elevated IgG 4 subclass but also found significant variability among studies. A total of 27 publications reported treating individuals with ASD using IVIG, including four prospective, controlled studies (one was a double-blind, placebo-controlled study); six prospective, uncontrolled studies; 2 retrospective, controlled studies; and 15 retrospective, uncontrolled studies. In some studies, clinical improvements were observed in communication, irritability, hyperactivity, cognition, attention, social interaction, eye contact, echolalia, speech, response to commands, drowsiness, decreased activity and in some cases, the complete resolution of ASD symptoms. Several studies reported some loss of these improvements when IVIG was stopped. Meta-analysis combining the aberrant behavior checklist outcome from two studies demonstrated that IVIG treatment was significantly associated with improvements in total aberrant behavior and irritability (with large effect sizes), and hyperactivity and social withdrawal (with medium effect sizes). Several studies reported improvements in pro-inflammatory cytokines (including TNF-alpha). Six studies reported improvements in seizures with IVIG (including patients with refractory seizures), with one study reporting a worsening of seizures when IVIG was stopped. Other studies demonstrated improvements in recurrent infections, appetite, weight gain, neuropathy, dysautonomia, and gastrointestinal symptoms. Adverse events were generally limited but included headaches, vomiting, worsening behaviors, anxiety, fever, nausea, fatigue, and rash. Many studies were limited by the lack of standardized objective outcome measures. IVIG is a promising and potentially effective treatment for symptoms in individuals with ASD; further research is needed to provide solid evidence of efficacy and determine the subset of children with ASD who may best respond to this treatment as well as to investigate biomarkers which might help identify responsive candidates.

## 1. Introduction

Autism spectrum disorder (ASD) is a neurodevelopmental disorder which is behaviorally defined by impairments in social communication and the presence of repetitive and restrictive behaviors. ASD affects approximately 2% of children in the United States [1]. Despite decades of research, the etiology and treatment of children with ASD is still very incomplete. This has resulted in a minority of children reaching optimal outcomes with many manifesting symptoms into adulthood and resulting in substantial economic and societal costs [2].

A number of medical comorbidities have been reported in ASD including mitochondrial dysfunction [3], sleep disorders [4], immune related problems [5], gastrointestinal abnormalities [6], inflammation [7], and epilepsy [8]. Addressing these comorbid conditions has the potential to improve the ability to function and the quality of life of children with ASD and their families [9]. One of the more recent promising areas of research is dysfunction of the immune system [10], which is a potential target for treatment.

Several lines of evidence link immune abnormalities to ASD. Family history of atopic [11] and autoimmune [12] disease is associated with ASD. Maternal immune activation during pregnancy has been shown to be associated with an increase in risk of ASD in the offspring in human [13] and animal studies [14]. Individuals with ASD demonstrate specific human leukocyte antigen risk alleles that put them at risk for immune dysfunction [15]. Elevations in specific monocyte cytokine profiles have been associated with ASD [16,17] and specific patterns of inflammatory cytokines have been identified in the cerebrospinal fluid and brain in individuals with ASD [18]. Autoantibodies to the brain [19] and other important proteins such as the folate receptor alpha [20] also appear to be associated with ASD.

Children with ASD also appear to be at increased risk for clinical immune disorders. Some of the immune-related problems reported in ASD include common variable immunodeficiency (CVID), hypogammaglobulinemia (i.e., low total Immunoglobulin G (IgG)) [21] and specific polysaccharide antibody deficiency (SPAD) [22]. One study reported that lower levels of IgG were associated with more severe aberrant behaviors in children with ASD [23]. A number of studies have reported on the use of intravenous immunoglobulin (IVIG) in ASD to treat immune-related problems [21]. Some of the medical comorbidities reported in ASD might also improve with the use of IVIG. For example, IVIG has been shown to have anti-seizure properties [24,25,26] and anti-inflammatory effects [27,28,29,30]. The anti-inflammatory effects of IVIG are observed at higher doses of IVIG (i.e., 2 grams/kg) for inflammatory and autoimmune disorders [31].

This paper is a systematic review and meta-analysis examining the evidence for abnormal IgG concentrations and the therapeutic use of IVIG in ASD. Adverse events (AEs) are also collated. This study demonstrates that IVIG is a promising and potentially effective treatment for symptoms in individuals with ASD, but further research is needed to provide solid evidence of efficacy and determine the subset of children with ASD who may best respond to this treatment, as well as to investigate biomarkers which might help identify responsive candidates.

## 2. Materials and Methods

### 2.1. Search Strategy

A prospective protocol for this systematic review was developed a priori, and the search terms and selection criteria were chosen in an attempt to capture all pertinent publications. A computer-aided search of PUBMED, Google Scholar, EmBase, Scopus and ERIC databases from inception through March 2021 was conducted to identify pertinent publications using the search terms ‘autism’, ‘autistic’, ‘Asperger’, ‘ASD’, ‘pervasive’, and ‘pervasive developmental disorder’ in all combinations with the terms ‘IVIG’, ‘IgG’, ‘immunoglobulin’, ‘immunoglobulins’, ‘globulin’, ‘intravenous immunoglobulin’, ‘gamma globulin’ and ‘immunodeficiency.’ The references cited in identified publications were also searched to locate additional studies. Appendix A depicts the publications identified during the search process.

### 2.2. Study Selection and Assessment

This systematic review and meta-analysis followed PRISMA guidelines [32]. The PRISMA Checklist and Flowchart can be found in Appendix A. One reviewer screened titles and abstracts of all potentially relevant publications. Studies were initially included if they (1) involved individuals with ASD; and (2) reported on the use of IVIG or reported IgG concentrations. Animal models were excluded. Abstracts or posters from conference proceedings were included if published in a journal. Articles were excluded if they: (1) Did not involve humans (for example, cellular models); or (2) did not present new or unique data (such as review articles or letters to the editor). After screening all records, 38 publications met inclusion criteria (see Appendix A); two reviewers then independently reviewed these articles for inclusion and assessed factors such as the risk of bias. As per standardized guidelines [33], selection, performance detection, attrition, and reporting biases were considered. One study reported on IgG levels and also treatment with IVIG. Two studies reported on the use of oral immunoglobulin in ASD and were not included in the analysis [34,35]. One manuscript reported an ongoing double-blind placebo controlled (DBPC) crossover study which was never published and not included in the analysis [36].

### 2.3. Meta-Analysis

MetaXL Version 5.2 (EpiGear International Pty Ltd., Sunrise Beach, Queensland, Australia) was used with Microsoft Excel Version 16.0.12827.20200 (Redmond, WA, USA) to perform the meta-analysis. Mean immunoglobulin titers were pooled across studies using standard methodology [37]. Various manuscripts reported immunoglobulin concentrations in different units. For consistency we report concentrations in mg/dL. In some papers the interquartile intervals were reported rather than standard deviations. In such cases we use the estimator for estimating the standard deviation from interquartile range as defined by the Cochrane Handbook [38]. The data from this meta-analysis is available upon request to the authors.

To compare immunoglobulin titers across groups, pooled mean differences were calculated using the inverse variance heterogeneity model since it has been shown to resolve issues with underestimation of the statistical error and spuriously overconfident estimates with the random effects model when analyzing continuous outcome measures [39]. Cochran’s Q was calculated to determine heterogeneity of effects across studies and when significant, the I^2^ statistic (Heterogeneity Index) was calculated to determine the percentage of variation across studies that is due to heterogeneity rather than chance [40,41]. The Luis Furuya-Kanamori (LFK) Index derived from Doi plots were reviewed for significant asymmetries (>±2) in the prevalence distribution when there were 3 or more studies [42,43].

Few intervention studies used quantitative standardized outcome measures, but two used the Social Responsiveness Scale (SRS) and five used the Aberrant Behavioral Checklist (ABC). Of note, the social withdrawal subscale of the ABC is called lethargy and inadequate eye contact in some studies, but we will refer to it as social withdrawal throughout to be consistent. Only two treatment studies contained enough details to be included in the treatment meta-analysis. Random-effects models, which assume variability in effects from both sampling error and study level differences [44,45], were used to calculate pooled standardized mean effect and pooled effect size. Effect sizes were considered small if Cohen’s d’ was 0.2; medium for Cohen’s d’ was 0.5; and large if Cohen’s d’ was 0.8 or higher [46].

## 3. Results

### 3.1. Studies of IgG Concentrations in Autism Spectrum Disorder

Articles examining IgG concentrations in individuals with ASD are first examined followed by studies which have reported therapeutic IVIG use in ASD.

#### 3.1.1. Studies on IgG Concentrations in ASD

Twelve studies were identified that measured IgG concentrations in children with ASD (Table 1). Eight studies examined serum IgG with six using controls and two using standard reference ranges. Two studies examined serum IgG in neonates with both using controls. Two studies examined IgG in cerebrospinal fluid (CSF) with one using controls and one using a standard reference range. Most studies only had a modest number of participants with only four studies having a relatively large number of participants (i.e., 80+).

Overall, of the eight studies that examined serum IgG in children with ASD, two reported higher and one reported lower total IgG while the others found no significant difference. One study found increased IgG2 subclass concentrations and three studies reported higher IgG4 concentrations. No studies reports lower IgG subclasses. However, one study found that lower total IgG concentrations was correlated with increased severity of disruptive behaviors as measured by the ABC [23], while another study found that higher IgG 4 concentrations were significantly associated with an increased severity of social interaction impairments as measured by the Autism Diagnostic Observation Schedule (ADOS) [49].

Two studies examined birth samples from archived newborn blood specimens which were obtained from the California Genetic Disease Screening Program and reported that ASD risk was associated with lower total IgG concentrations in the neonatal period [54,55].

The two studies that examined CSF IgG found no significant difference in CSF IgG concentrations or the IgG index as compared to the reference groups [56,57]. However, one study did find increased protein in 33% of ASD cases and oligoclonal bands in one patient (3%) and GAD65 antibodies in two patients (6%) in the CSF.

#### 3.1.2. Meta-Analysis of Immunoglobulin G Concentrations in ASD

Nine studies used control groups as comparisons with two large studies using neonates while the remainder examined children. Seven studies used TD unrelated controls, while two studies used TD sibling controls and three studies used unrelated DD controls. One study divided the ASD groups into those with autistic disorder and those without autistic disorder but with autism features. Because it is not clear how this latter group would map onto the current definition of ASD, it has not been included in the analysis. One of the two studies examining neonates did not provide descriptive statistics of the immunoglobulin concentrations so the studies could not be combined. Only one study used parents as controls and the two studies that used developmentally delayed control groups studied different immunoglobulin measures, so parents and developmentally delayed controls were not included in the meta-analysis.

Table 2 outlines the pooled mean difference when combining studies using non-sibling and sibling controls separately as well as all studies combined (see Appendix A for the number of participants per comparison). Both the pooled mean difference for the non-sibling and combined studies found that total IgG was lower in the ASD group with a confidence interval that did not include zero. However, the meta-analysis models for these pooled mean differences were not significant because of large variation among studies, as manifested by the significant Cochran’s Q and DOI plots showing major asymmetry with LFK indexes of 8.80 and 8.99, respectively. Pooled mean difference for IgG 4 subclass was significantly elevated in individuals with ASD for all comparisons, with the comparison between ASD and TD siblings demonstrating significant variability across studies. None of the other IgG subclasses were found to be significantly different between ASD and comparisons groups in the meta-analysis.

#### 3.1.3. Summary of Immunoglobulin G Concentrations in ASD

Overall, studies on IgG abnormalities in ASD suggest that individuals with ASD may have a wider variation in serum IgG concentrations as compared to non-ASD controls. There appears to be preliminary evidence for both depressed total IgG and elevated IgG 4 subclass, both of which appear to have concentrations related to symptomatology. This may suggest two different immune abnormalities in different subsets of patients. However, with only a few large well-controlled studies, it is difficult to make any firm conclusions.

### 3.2. The Theraputic Use of IVIG in ASD

A total of 27 publications were identified which examined the use of IVIG in ASD. Four studies were prospective, controlled studies; six were prospective, uncontrolled studies; 2 were retrospective, controlled studies; and 15 were retrospective, uncontrolled studies (case reports and series).

#### 3.2.1. Prospective, Controlled Studies

Four prospective, controlled studies were identified (Table 3). One study used IVIG in children without immune related problems [58] while the children in the other three studies had immune related problems.

The first study was a DBPC crossover study of a one-time dose of IVIG 0.4g/kg or placebo in 12 children with ASD (age range 4.2 to 14.9 years) and reported significant improvements as rated by parents and teachers in ABC irritability, hyperactivity, social withdrawal and inappropriate speech. Improvements were also observed in drowsiness and decreased activity on the Symptom Checklist compared to the placebo group. Significant improvements were not observed by physicians as rated by the Children’s Psychiatric Rating Scale (CPRS). Of note, none of the ASD children in this study had abnormalities in IgG or IgM concentrations [58].

Three other studies contained control groups who did not receive a placebo. In the first, 10 children with ASD (age not noted) with specific polysaccharide antibody deficiency (SPAD) and hypogammaglobulinemia were treated with an unspecified IVIG dose and compared to 14 non-ASD children with similar immunodeficiency treated with IVIG and 49 ASD and 39 normal children who did not receive treatment. Pro-inflammatory cytokines (IL-6, IL-12 and IL-23) and productions of IL-12 with exposure to phytohemagglutinin/Concanavalin A and IL-17/IFN-γ with exposure to phytohemagglutinin decreased, while TGF-ß and sTNFR II increased in the children with ASD who received IVIG compared to normal controls. ASD behaviors were not reported in this study [22].

In the second study, seven children with ASD and immunodeficiency (one with CVID and six with SPAD) were treated with any unspecified dose of IVIG, but no effects of IVIG were discussed [59].

Finally, in a third study, 78 children with ASD (ages 2–10 years old, 47 boys, 31 girls) were treated with IVIG 2 g/kg per month for six months. Characteristics of the treated children were compared to characteristics of a control group of 32 ASD children who received conventional therapy without IVIG. Additionally, changes in the ABC scale were compared to baseline measurements in the treatment group. Inclusion criteria for IVIG treatment included the presence of two to four polymorphisms in folate cycle genes, deficiency in natural killer cells, reactivated herpes and/or measles virus infections, or signs of leukoencephalopathy on a brain MRI. The authors reported “complete elimination of the phenotype of autism spectrum disorders” in 21 (27%), “marked improvement” in 33 (42) and mild-to-moderate improvements in 23 (29%). Overall, 77 (99%) had some improvement with IVIG. In the 21 children with the most improvements, one (5%) lost improvements when IVIG was stopped and 12 (50%) who showed mild-to-moderate improvement lost their gains two to four months after completing therapy. Twenty-nine out of 36 patients (81%) with epilepsy had improvement in seizures and 49 out of 68 children (72%) had improvements in gastrointestinal symptoms [60]. Compared to baseline, the treated group improved in ABC irritability, hyperactivity, social withdrawal and inappropriate speech.

#### 3.2.2. Prospective, Uncontrolled Studies

Six prospective, uncontrolled studies were identified (Table 4); five studies administered IVIG to individuals with ASD who had immune related abnormalities whereas one study used IVIG in patients without immune problems [61].

In a prospective, open-label study, 10 children (ages three to six years old) with ASD and IgG deficiency or other immune abnormalities were treated with IVIG (0.4 g/kg every four weeks for six months). Four (40%) showed “marked improvements”, one (10%) showed “striking improvements” and five (50%) had “minimal improvements.” Improvements were noted in social interaction, eye contact, echolalia and better response to commands. Speech improvements included better articulation and improved vocabulary. Younger children had more improvements compared to older patients [52].

In another prospective, open-label study, 10 children with ASD without immune problems (four to 15 years old, mean age eight years) received 0.154–0.375 g/kg IVIG (mean dose 0.27 g/kg) every six weeks. Six children received four infusions, whereas the other four received one, three, five, and six infusions, respectively. Five (50%) did not show improvements, while four (40%) had mild improvements in attention and hyperactivity reported by parents but not confirmed by clinicians; parents of these four children decided not to continue IVIG due to the cost and inconvenience. One child (10%) who received 0.375 g/kg for four infusions had “a very significant amelioration of autistic symptoms”; when the treatment was stopped the improvements were lost over a three-month period [61].

In another prospective, open-label study, seven children with ASD (ages three to six years old) with a history of recurrent infections or seizures were treated with 0.4 g/kg of IVIG every month for six months, with five children completing the study. Children were evaluated with Ritvo-Freeman Real Life Rating Scale (RF), Children Yale-Brown Obsessive-Compulsive Scale, Clinical Global Impression Scale (CGI), and the Autism Modification of the NIMH Global Obsessive-Compulsive Scale. Improvement was noted on the RF sensory responses but was considered not significant after Bonferroni correction [62].

In another prospective, open-label study, 27 children with ASD (ages two to 10 years old, median three years) and immune abnormalities were treated with IVIG 0.4–1.0g/kg every three weeks for six to 18 months. Immune abnormalities included IgG deficiency in five (19%), IgG subclass deficiency in 12 (44%), or recurrent infections that did not respond to conventional therapy with the presence of functional antibody deficiency in 10 (37%). With IVIG treatment, improvements in infections were observed in otitis media in 19 (70%), upper respiratory infections in 11 (41%), and sinopulmonary infections in nine (33%). Parents and physicians reported improvements in ASD symptoms in 21 (78%) [63].

In another prospective study, 12 children with ASD (age not reported) with either a humoral or cellular immune deficit (or both) were treated with 1.0 g/kg of IVIG monthly for three years. All children had a drastic reduction in the number of infections and improvements in cognition, communication, verbal interaction and social skills (1 to 4 point improvements on a scale of 1 to 5) [64].

In a prospective, open-label 30-week study, 14 children with ASD (mean age 7.6 ± 3.0 years) and immune abnormalities received 1 g/kg of IVIG every two to four weeks for 10 treatments. Immune abnormalities include T or B cell dysfunction, activated CD154 levels <80, or abnormal lymphocyte stimulation test and/or recurrent infections. Outcomes were compared to baseline measurements. Significant improvements were observed in CGI-S total score, CGI-I total score, and SRS total score. Significant improvements on Children’s Communication Checklist–2 speech and semantics as well as Autism Diagnostic Observation Schedule (ADOS) stereotyped behaviors and restricted interest, communication plus social interaction total, and reciprocal social interaction were found. Statistically significant improvement on the ABC was only found on the hyperactivity subscale. Significant decreases in TNF-α induced by TLR ligands zymosan, flagellin, and lipopolysaccharide were also reported [65].

#### 3.2.3. Retrospective, Baseline Controlled Case Series with Prospectively Collected Outcomes

Two case series (Table 5) were identified and are reviewed by year published. All studies used IVIG in ASD patients with immune-related problems.

In a retrospective case series, 26 children with ASD (ages three to 17 years, mean age 6.8 years) and neurodevelopmental regression (mean age 17 months) were treated with 0.4 g/kg of IVIG monthly for six months. Only 23% of treated patients had low immunoglobulin levels. Total ABC score as well as hyperactivity, inappropriate speech, irritability, social withdrawal and stereotypy subscales showed significant improvement. Twenty-two (85%) lost some improvements when IVIG was stopped [66].

In an open-label study, 31 children with ASD who had evidence of autoimmune encephalopathy (presence of various brain autoantibodies) were treated with 0.75–2g/kg IVIG every two to six weeks; 77% were treated for more than one year. Significant improvements were observed in SRS total score and cognition and mannerisms subscales and ABC total score and irritability, lethargy, hyperactivity and inappropriate speech subscales [19]. Of interest, this study also found that the anti-Dopamine D2L and anti-tubulin antibodies of the Cunningham panel (Moleculera, Oklahoma City, OK, USA) were predictive of response to IVIG.

#### 3.2.4. Retrospective, Uncontrolled Case Series

Three case series (Table 6) were identified and are reviewed by year published. All studies used IVIG in ASD patients with immune-related problems.

Three children with ASD and intractable epilepsy defined as at least three seizures per day were treated with 1.0–1.7 g/kg of IVIG for up to 11 months. One child had low total IgG and the others had normal IgG levels. Two patients became seizure-free and the third had a reduction in seizures. Two children had improvement in social interaction, communication and behavior. In one child, seizures and ASD symptoms worsened when IVIG was discontinued [67].

Eight children with ASD and SPAD (ages six to 16 years old) who had worsening of cognitive skills and behavioral symptoms with “immune insults” were treated with 0.6–1 g/kg of IVIG every three weeks for one to six years. Three children had below normal total IgG. Treatment improved recurrent infections. In the four patients with treatment resistant epilepsy, seizures were significantly improved. Overall ASD symptoms were not reported to improve but behavioral exacerbations associated with infections did improve. Parents reported improvement in cognitive skills in one child but standardized clinical outcome measures did not confirm this [68].

Finally, in a retrospective case series, three children with ASD who had IgG deficiency were treated with an unspecified dose of IVIG over five to eight years. Serum IgG and IgM concentrations improved with improvement in immunoglobulin concentrations believed to correlated with an improvement in ASD symptoms [69].

#### 3.2.5. Retrospective, Uncontrolled Case Reports

Twelve studies were case reports (Table 7) and described patients with ASD treated with IVIG for mostly immune-mediated conditions, with two publications reporting on the same case [70,71]. Indications for IVIG ranged from purely immune conditions to immune related neurological conditions, as well as one case report for neuroleptic malignant syndrome which is not usually believe to be an immune related condition.

Three case reports describe individuals with ASD and immune conditions. A 15-year-old boy with ASD and CVID treated with IVIG was reported to have resolution of recurrent infections and marked improvement in ASD symptoms [72]. A 22-year-old man with ASD with a large mesenteric granulomatous mass and CVID was treated with monthly IVIG. Appetite, weight and serious infections improved but no changes in ASD behaviors were mentioned [73]. Finally, a 13-year-old boy with ASD and CVID with recurrent infections was treated with IVIG but he continued to have infections and no note was made of changes in ASD behaviors [74].

Two cases of patients with inflammatory neuropathies were reported. An eight-year-old boy with ASD, epilepsy and chronic inflammatory demyelinating polyneuropathy (CIDP) was treated with IVIG and IV methylprednisolone, but weakness progressed. No mention was made of any changes in ASD symptoms with IVIG [75]. A six-year-old girl with ASD and right upper arm weakness due to a demyelinating neuropathy was treated with one course of IVIG with resolution of the neuropathy symptoms but no mention if ASD symptoms improved [76].

Five cases of immune mediated encephalopathy were reported with four of the cases being autoimmune encephalopathy. A 33-month-old boy with ASD and neurodevelopmental regression was diagnosed with anti-N-Methyl-D-aspartic acid (NMDA) receptor encephalitis by lumbar puncture. Treatment with 2 g/kg of IVIG over five days resulted in improvements in language and social skills on the third treatment day [77]. A five-year-old ASD girl diagnosed with anti-NMDA receptor encephalitis by lumbar puncture was treated with 20 mg/kg/day of IV steroids for five days, 0.4 g/kg of IVIG for five days and 0.375 g/kg of rituximab weekly for four weeks. Improvements were noted in seizures and language with this combination [78]. A five-year-old boy experiencing neurodevelopmental regression with the emergence of ASD symptoms including social isolation and repetitive behaviors was diagnosed with autoimmune encephalitis due to alpha-3 subunit nicotinic acetylcholine receptor autoantibodies. Treatment with plasmapheresis followed by monthly IVIG resulted in significant improvements after five months with the child being able to attend a regular classroom, resolution of hyperactivity and improvements in social interaction [79]. A five-year-old ASD boy with infection-induced autoimmune encephalopathy was treated with 1.6 g/kg of IVIG every eight weeks for two years, resulting in marked improvements in social interaction, language, learning and memory; he maintained the cognitive improvements when IVIG was stopped [80]. Lastly, a 14-year-old previously healthy girl with neurodevelopmental regression and the development of ASD symptoms including lack of eye contact, impaired communication, and social withdrawal with the onset of seizures was found to have enterovirus encephalitis as well as a low IgG index in the CSF. Treatment with 20 mg/kg of IVIG for five days resulted in improvements in eye contact, speech, and communication, as well as a resolution of her seizures [70,71].

Finally, a 32-year-old man with ASD and mental retardation diagnosed with neuroleptic malignant syndrome and seizures was treated with IVIG after he failed multiple anti-epileptics. Two weeks later, his seizures and dysautonomia resolved; no mention was made if ASD symptoms improved [81].

#### 3.2.6. Meta-Analysis of Behavioral Responses to Intravenous Immunoglobulin

Five studies used the ABC instrument as an outcome measure, but only two studies provided information to obtain means and standard deviations before and after the treatments to combine outcome measures. Both retrospective studies obtained data prospectively with baseline measurements [19,66]. Two other studies also observed changes in ABC scores from baseline but one did not report any variation measure [60] and the other reported range as a measure of variation [65]. The DBPC crossover study [58] is not appropriate to combine with the open-label studies because of the differences in methodology. Thus, two studies with a total of 46 participants with two sets of measurements for each participant (baseline and treatment) were included in the meta-analysis.

ABC irritability and total score were found to significantly improve with IVIG with a large effect size [total: d’ = 0.80 (0.37, 1.23), *p* < 0.001; irritability: d’ = 0.87 (0.44, 1.31), *p* < 0.0001]. ABC social Withdrawal and hyperactivity were found to significantly improve with IVIG treatment with a medium effect size [social withdrawal: d’ = 0.54 (0.12, 0.95), *p* = 0.01; hyperactivity: d’ = 0.67 (0.25, 1.09), *p* = 0.001]. ABC stereotyped movements and inappropriate speech were not found to significantly improve when the studies were combined.

Two studies [19,65] used the SRS as an outcome measure but one study reported range as a variation measure [65], so these studies could not be combined.

### 3.3. Adverse Effects

Most studies did not report any AEs. One case series specifically reported that no AEs were observed. None of the prospective controlled studies reported AEs. Two prospective uncontrolled studies reported AEs. Melamed et al. (2018) reported that four (29%) had infusion site reactions and three (21%) had a headache. Connery et al. (2018) reported AEs, mostly during the infusion, with headaches in 39%, vomiting in 29%, worsening behaviors in 16%, anxiety in 13%, fever in 13%, nausea in 10%, fatigue in 10%, and rash in 5%; two (6%) patients discontinued IVIG because of AEs. One case report reported agitation, combative behavior, and fearfulness [80]. None of the other case reports reported AEs. Thus, the majority of studies did not report AEs, but most studies were not specifically designed to follow AEs as outcome measurements and none of the studies were designed to measure safety. As most of the AEs of IVIG are well known, many AEs may have been considered part of the standard treatment. Most of the AEs reported are consistent with standard IVIG treatment.

## 4. Discussion

This systemic review aimed to identify studies examining IgG concentrations and the use of IVIG in individuals with ASD. We identified 12 studies which examined IgG concentrations and 27 studies that described the use of IVIG in ASD (one study fell into both categories). We found limited evidence for changes in immunoglobulin concentrations in children with ASD, suggesting this might be present in subgroups of children with more severe ASD. Evidence for the effectiveness of IVIG treatment was also found but studies demonstrated many limitations. In most cases, IVIG was used to treat immune abnormalities in individuals with ASD as IVIG is a common, safe and well-tolerated treatment for immune disorders.

Nine of the 12 studies examining IgG concentrations included controls, with two using sibling controls and seven using non-related controls; one study used parents as comparisons and three used non-ASD developmentally delayed controls as a comparison group. One study found that total IgG and IgG4 concentrations were higher in ASD as compared to sibling controls, while the other sibling study demonstrated no differences between those with and without ASD. In studies with non-related control children, two studies demonstrated elevated IgG and IgG4 concentrations in ASD as compared to controls, with one study finding that higher IgG4 concentrations were associated with more severe social interaction impairments in those with ASD. Three studies demonstrated lower IgG in ASD as compared to non-related controls; one study demonstrated a significantly lowered median IgG concentration in ASD as compared to controls, with the depression in IgG concentration related to more severe aberrant behaviors. One study demonstrated that 20% of children with ASD had subclass deficiency, including two with IgG4 deficiency, and another study demonstrated a lower IgG concentration in the neonatal period for those who went on to be diagnosed with ASD, with neonatal IgG concentration significantly related to the risk of developing ASD. Two studies demonstrated no differences in serum IgG concentrations between ASD and control participants, while another study demonstrated no difference in the CSF IgG index between ASD and control individuals.

Overall, immunoglobulin studies in ASD reveal two patterns of IgG alterations. Three studies demonstrate depressed total IgG levels and three studies demonstrated elevated IgG concentrations which were primary driven by elevated IgG4 concentrations. Interesting, both abnormalities were correlated with more severe ASD-related symptoms. The meta-analysis was consistent with this pattern but also demonstrated significant variability among studies which would be expected if there are subgroups of children with different immunological profiles.

Meta-analysis demonstrated a trend toward lower total IgG concentrations in children with ASD as a group with a large variation in this finding across studies. Several lines of evidence support the notion that this finding is driven by a subgroup of children with ASD and hypogammaglobulinemia rather than simple biological variability. First, a lower total IgG concentration has been found to be correlated with more severe aberrant behaviors in the only large study to examine this relationship [23]. Second, studies on neonates suggest that lower IgG levels were associated with an increased risk for developing ASD [54,55]. Third, treatment studies have suggested that treatment of children with ASD and hypogammaglobulinemia results in improved ASD symptoms [52,63,64,66,67,69].

The fact that low total IgG levels in the neonatal period is associated with the development of ASD may suggest that the humoral immune system may be depressed early in life. Consistent with humoral immune system abnormalities, children with ASD are more likely to have recurrent viral and bacterial infections. For example, a history of otitis media (OM) and antibiotic use is associated with increased risk of ASD [82]; children with ASD are more likely to have OM, especially complicated OM, suggesting more severe OM infections [83], as compared to non-ASD siblings; children with ASD are more likely to have recurrent OM and upper respiratory and other infections [5,84]; and those children with ASD who have recurrent infections are found to be more medically complex and lower functioning [84]. Recurrent infections could be linked to depressed immune response. Studies have suggested that children with ASD and recurrent infections and other immune abnormalities have associated abnormal Toll-like receptor responses [85], dysregulation of inflammatory and counterregulatory cytokines [86], changes in regulatory microRNA [17], and atypical mitochondrial respiration [16].

Meta-analysis has demonstrated that elevation in IgG 4 subclass is related to ASD. Although commonly associated IgG4-related disease, such as fibro-inflammatory changes in the salivary and lacrimal glands, orbit, pancreas, and kidneys [87] is not common for children with ASD, elevated IgG4 is associated with eosinophilic esophagitis [88], a disorder that is under diagnosed and associated with restricted feeding in ASD [89]. This raises the possibility that IgG4 could potentially be used to help differentiate children with feeding disorders due to eosinophilic esophagitis and behavioral issues. Clearly this is an important and promising area of research.

Interestingly, eosinophilic esophagitis is associated with IgE-mediated food allergies which is associated with enteric microbiome alterations in non-ASD children [90,91]. As microbiome imbalances have been associated with ASD [92] and may have important consequences in immune regulation [93], microbiome alterations may be playing a role in immune dysregulation in ASD. The fact that chronic gastrointestinal symptoms have also been linked to non-IgE-mediated food allergies [94] and immune dysregulation [85] in children with ASD demonstrates the complex relationship between gastrointestinal symptoms and immune dysfunction in ASD and the complicated nature of management and treatment of ongoing symptoms in ASD.

Studies which have been conducted on measuring IgG in ASD have been innovative in examining IgG concentrations prior to symptom onset to help understand the role of immune dysfunction in the etiology of ASD and in examining their parents given the research that suggests a transgenerational effect in the etiology of ASD. Clearly, further studies will be needed to better understand the potential subgroups of children with ASD and immune abnormalities, especially with respect to the developmental nature of IgG concentrations. These studies point to the possibility of subsets of children with ASD with different immune profiles, but further studies are needed to verify these abnormalities.

There were four prospective controlled studies of IVIG but only one of these was a DBPC study [58] and only two examined changes in ASD symptoms, with both studies reporting improvements. One of these aforementioned studies enrolled children with immune abnormalities while the other did not. One of the other two studies which did not examine changes in symptoms instead measured changes in cytokines with IVIG. The limited number of controlled and placebo-controlled studies is disappointing but given the risk-benefit of an intravenous treatment in children, a high level of certainty of the efficacy is needed before such studies are launched, so the lack of such studies is understandable at this time. More studies will be necessary for future evaluation of this treatment.

Six studies were prospective without a control group with five of the studies enrolling children with ASD and immune problems. Immune problems varied widely from recurrent infections to quantitatively diagnosed immune deficiencies. All studies reported improvements but in one study the improvements were not statistically significant after correction for multiple comparisons. Unfortunately, only two of the six studies used standardized clinical outcome measures to document improvements.

Two studies were retrospective case series of prospectively collected outcomes with both studies using the ABC questionnaire, allowing the combining of these studies in a meta-analysis, which demonstrated improvements on the ABC scale including significant improvements in irritability, hyperactivity, and social withdrawal, as well as total aberrant behaviors (all with medium to large effect sizes).

Three case series and twelve case reports described treatment of children with ASD with IVIG. Only one study used standardized quantitative clinical outcome measures and all studies which reported changes in ASD symptoms reported improvements. Only one case report and one case series enrolled patients without immune-related problems.

One of the major limitations of the IVIG treatment studies was the lack of standardized outcome measures. The most commonly used standardized outcome measure was the ABC, which was used in five studies, all of which demonstrated improvement in aberrant behaviors. The next most commonly used standardized outcome measure was the SRS in two studies, both of which demonstrated improvements in social function. Other studies noted clinical improvements in communication, irritability, hyperactivity, cognition, attention, social interaction, eye contact, echolalia, speech, and responsiveness, although standardized measures were not always used to collect these observations. Besides improvements in ASD symptoms, other benefits were seen from IVIG. Six studies reported improvements in seizures with IVIG [60,67,68,76,78,81], with one study reporting a worsening of seizures when IVIG was stopped [67]. Other studies demonstrated improvements in recurrent infections [64,73], appetite [73], weight gain [73], neuropathy [76], dysautonomia [81], pro-inflammatory cytokines [22], and gastrointestinal symptoms [60].

The dose of IVIG varied widely from 20 mg/kg to 2 g/kg and the treatment duration varied from one total IVIG treatment to recurrent treatment for many years, but most studies did not document the exact dose and frequency of the treatment. Overall, most studies did not report if AEs occurred, and for the four studies that did, the AEs were of limited duration and severity [19,65,67,80].

The biological mechanism by which IVIG has its therapeutic effect was investigated in a limited number of studies. One prospective controlled study [22] and one prospective uncontrolled study [65] found a decrease in inflammatory cytokines, and one case series demonstrated an improvement in IgG and IgM concentrations [69] with IVIG treatment. Given that IgG and IgM [23] and inflammatory cytokine [17] concentrations have been associated with ASD behaviors, modulation of these factors could play a role in the therapeutic effect of IVIG. Two prospective uncontrolled studies [63,64] and two case series [72,73] noted a substantial decrease in the number of infections with IVIG treatment, and one case series noted an improvement in behavioral exacerbations associated with infections [68]. Given that a subgroup of children with ASD have been described with immune abnormalities who have behavioral exacerbation with infections [86], simply improving the number of infections may improve the ability to function and quality of life for some individuals with ASD. The therapeutic effect of IVIG for modulating autoantibodies may have been effective in the case-series [19] and case-studies [70,71,77,78,79,80] of patients with ASD and autoimmune encephalopathy caused by anti-NMDA receptor [77,78], anti-nAChR receptor [79], anti-dopamine receptor [19], and anti-tubulin [19] autoantibodies. Finally, the therapeutic effect of IVIG on seizures may have been therapeutic in the cases described with refractory seizures [67,68,70,71].

Thus, these studies suggest IVIG treatment can be effective for some individuals with ASD, particularly those with underlying immune-related problems. However, many studies had substantial limitations, including a lack of a control group, only qualitative outcomes reported, no standardized reporting of details of dosing and duration of treatment, and no standardized reporting of AEs. Indeed, this does suggest that the majority of the studies are open to bias. Additionally, several studies did not report commonly used measures of variance such as standard deviation or interquartile range, making their data difficult to include in a meta-analysis. Thus, future studies will need to address these limitations in order to provide high quality evidence for the use of IVIG in ASD.

## 5. Conclusions

ASD is a prevalent and life-long neurodevelopmental disorder with no known cure. Standard of care treatments are effective in some individuals but leave many with incomplete recovery. A better understanding of the underlying physiological abnormalities is beginning to emerge with evidence supporting abnormalities in immune function, making the immune system a potential target for treatment. Several common treatments for children with ASD also have effects on the immune system, suggesting that their efficacy in ASD may in part be linked to their effects on modulating immune function [95]. This review provide support for the notion that at least a subset of children with ASD have immune abnormalities, particularly in humoral immunity characterized by abnormal concentrations of immunoglobulins and may respond to the immune modulating effect of IVIG therapy. This study has found that there is limited evidence that some children with ASD have abnormal IgG concentrations, but this may be driven by a subgroup with abnormalities. The other major finding in this meta-analysis was an elevation in IgG4 subclass. Variations in IgG concentration were perhaps related to ASD symptom severity, suggesting that for some individuals with abnormal IgG concentrations, IVIG may be a directly targeted treatment.

Still IVIG has many other clinical effects aside from replacing endogenous IgG. Indeed, IVIG can modulate the immune system and is commonly used in neurological disorders to treat pathophysiological processes involving inappropriate activation of the immune system. IVIG is a common treatment because it is usually well tolerated with minimal and non-serious AEs. Additionally, as compared to other treatments which modulate the immune system, IVIG modulates immune function while also providing immune protection, so concerns for immune suppression and opportunistic infections are minimized.

Overall, IVIG appears to be effective in many children with ASD, particularly in those with identified immune problems. It also appears to be well tolerated. However, the quality of the evidence for the use of IVIG is still below what is commonly accepted for a routinely used treatment with the bulk of the studies being uncontrolled. Many studies demonstrated bias, including selection bias (lack of randomization), performance bias (lack of blinding), detection bias (lack of standardized outcomes), attrition bias (retrospective studies are prone to losing patients to follow-up), and reporting bias (case studies tend to report positive rather than negative outcomes). Thus, the current set of studies presented should be used to design and implement well-controlled, blinded randomized clinical trials in the future. Additionally, the populations used in these studies are very heterogeneous with many different immune system abnormalities, making it hard to determine if there is a particular subset of children with ASD in which the treatment may be most effective. Thus, further identification of biomarkers that can guide treatment will be helpful.

## 6. Patents

No patents to report.

## Figures and Tables

**Table 1 jpm-11-00488-t001:** Studies of Immunoglobulin G Concentration in Autism Spectrum Disorder. DD = Developmental Delay, TD = Typical Developing, P = Prospective, R = Retrospective; CSF = Cerebrospinal Fluid; AD = Autistic Disorder; NDR = Neurodevelopmental Regression.

Study	Study Type	Autism Group	Control Group	Outcomes
Studies in Children Using Contemporaneous Control Groups for Comparison				
Croonenberghs et al., 2002 [47]	P	18	22 TD	Total IgG, IgG2 and IgG4 higher in ASD No Difference in IgG1 and IgG3
Trajkovski et al., 2004 [48]	R	35	21 TD Siblings	Total IgG and IgG4 higher in ASD No Difference in IgG1, IgG2 and IgG3
Heuer et al., 2008 [23]	P	166 with AD 27 with ASD	96 TD 32 DD	Total IgG lower in AD Total IgG Inversely Correlated with Behavior
Enstrom et al., 2009 [49]	P	114	96 TD 31 DD	IgG4 higher in ASD No Difference in IgG1, IgG2 and IgG3 IgG4 Correlated with Social Impairment
Spiroski et al., 2009 [50]	R	30	22 TD Sibs 30 Moms 26 Dads	No Difference in Total IgG, IgG1, IgG2, IgG3 or IgG4 between ASD and TD Siblings
Wasilewska et al., 2012 [51]	P	24 NDR	14 TD	No Difference in Total IgG
Studies in Children Using Standard Reference Range as Comparison				
Gupta et al., 1996 [52]	P	25	Standard Reference	20% of ASD had below normal IgG subclasses (IgG1 in 1; IgG2 in 1; IgG3 in 1; IgG4 in 2)
Stern et al., 2005 [53]	P	24 Recurrent Infections	Standard Reference	No Difference in Total IgG
Studies in Neonates Using Contemporaneous Control Groups for Comparison				
Grether et al., 2010 [54]	R	213	265 TD	Neonatal Total IgG lower in ASD Lower IgG Associated with Increased ASD Risk
Grether et al., 2016 [55]	R	84	159 TD 49 DD	Lower IgG Associated with Increased ASD Risk
Studies on Immunoglobulin G Concentrations in the Cerebrospinal Fluid				
Young et al., 1977 [56]	P	5	Standard Reference	IgG in the CSF was normal
Runge et al., 2020 [57]	R	35	39 TD	No Difference in CSF IgG Index


**Table 2 jpm-11-00488-t002:** Meta-analysis of Studies on Immunoglobulin G Concentration in Autism Spectrum Disorder. Pooled mean difference (MD) with 95% confidence interval, Cochran’s Q (Q), Heterogeneity Index (I^2^). Statistics are estimated by inverse variance heterogeneity model. Significant values are Bold. * *p* ≤ 0.01; ** *p* ≤ 0.001.

	Non-Siblings			Siblings			All Controls		
	Pooled MD	Cochran’s Q	I^2^	Pooled MD	Cochran’s Q	I^2^	Pooled MD	Cochran’s Q	I^2^
Total IgG	−231 (−223, −238)	**40 ****	95%	49 (−3, 101)	3.3	70%	−225 (−217, –233)	**153 ****	97%
IgG1	14 (−45, 74)	2.2	54%	17 (−17, 51)	5.5	82%	17 (−13, 46)	7.7	61%
IgG2	9.2 (−1.0, 19.2)	0.0	0%	35.8 (3.5, 68.2)	5.8	83%	11.5 (1.9, 21.2)	8.23	64%
IgG3	-0.3 (−3.2, 2.6)	0.4	0%	0.5 (−3.7, 4.7)	0.4	0%	0.0 (−2.4, 2.4)	0.9	0%
IgG4	***16.6 ***** ***(6.7, 26.4)***	0.7	0%	***19.7 ***** ***(12.8, 26.5)***	**6.5 ***	84%	***18.7 ***** ***(13.1, 24.3)***	7.4	60%

**Table 3 jpm-11-00488-t003:** Prospective Controlled Immunoglobulin G Treatment Studies in Autism Spectrum Disorder. Specific Polysaccharide Antibody Deficiency (SPAD); Common Variable Immunodeficiency (CVID); Aberrant Behavior Checklist (ABC); Not specified (NS).

Study	Medical Indication	Autism Group (N)	IVIG Treatment	Outcomes
Niederhofer, Staffen et al., 2003 [58]	NS	12	400 mg/kg	Improvement in all ABC Subscales and improved drowsiness and activity
Jyonouchi, Geng et al., 2011 [22]	SPAD	10	NR	Decreased pro-inflammatory cytokines (IL-6, IL-12 and IL-23) and increased TGF-ß and sTNFRII
Jyonouchi, Geng et al., 2011 [59]	SPAD in 6 CVID in 1	7	NR	NR
Maltsev and Yevtushenko 2016 [60]	NK Cell Deficiency; Reactivated HSV or Measles Infection	78	2g/kg monthly for 6 months	Improvement in all ABC Subscales

**Table 4 jpm-11-00488-t004:** Prospective Uncontrolled Immunoglobulin G Treatment Studies in Autism Spectrum Disorder. Specific Polysaccharide Antibody Deficiency (SPAD); Common Variable Immunodeficiency (CVID); Aberrant Behavior Checklist (ABC); Clinical Global Impression Severity (CGI-S); Clinical Global Impression Improvement (CGI-I); Social Responsiveness Scale (SRS); Children’s Communication Checklist–2 (CCC-2); Autism Diagnostic Observation Scheduled (ADOS).

Study	Medical Indication	Autism Group	IVIG Treatment	Outcomes
Gupta et al., 1996 [52]	IgG deficiency and others	10	0.4 g/kg every 4 weeks for 6 months	No quantitative outcomes 5 with “marked” or “striking” improvements. 5 with minimal improvements
Plioplys 1998 [61]	None	10	154–375 mg/kg every 6 weeks for 1–6 infusions	No quantitative outcomes 1 remarkable, 4 mild and 5 no improvements
DelGiudice-Asch et al., 1999 [62]	Recurrent infections	7	400 mg/kg every month for 6 months	2 discontinued treatment No significant changes in several quantitative symptom scales
Oleske 2004 [63]	Antibody deficiency	27	0.4–1 g/kg every 3 weeks for 6–18 months	No quantitative outcomes Less recurrent infections ASD symptoms improved in 78%
Melamed et al., 2006 [64]	Humoral and/or cellular immune deficit	12	1 g/kg monthly for 3 years	Non-standard quantitative outcomes Drastic reduction in Infections Improvement in cognition, communication and social skills
Melamed et al., 2018 [65]	Activated CD154 levels <80, recurrent infections or abnormal lymphocyte stimulation test or	14	1 g/kg every 2–4 weeks for10 doses	Improvement in CGI-S and CGI-S, SRS, CCC–2 and ADOS

**Table 5 jpm-11-00488-t005:** Retrospective Case Series Immunoglobulin G Treatment Studies in Autism Spectrum Disorder with Prospectively Baseline Controlled Outcome Measures. Aberrant Behavior Checklist (ABC).

Study	Medical Indication	Autism Group (N)	IVIG Treatment	Outcomes
Boris et al., 2005 [66]	6 with IgG Deficiency	26	400 mg/kg every month for 6 months	Improvements in ABC total and all subscales 22 (85%) lost gains after stopping IVIG
Connery et al., 2018 [19]	Autoimmune encephalopathy	31	0.75–2 g/kg every 2–6 weeks; 77% treated >1 year	Improvements on SRS and ABC scales

**Table 6 jpm-11-00488-t006:** Retrospective Uncontrolled Case Series. Specific Polysaccharide Antibody Deficiency (SPAD); Aberrant Behavior Checklist (ABC).

Study	Medical Indication	Autism Group (N)	IVIG Treatment	Outcomes
Knutsen and Fenton 1998 [67]	1 with IgG deficiency 3 with intractable epilepsy	3	1.0–1.7 g/kg up to 11 months	No quantitative outcomes 2 seizure free, 1 with seizure improved ASD symptoms improved in 2 and worsened in 1
Jyonouchi et al., 2012 [68]	SPAD 4 with intractable epilepsy	8	0.6–1g/kg every 3 weeks for 1–6 years	No quantitative outcomes Four with seizure improvement One with cognitive improvements
Fadeyi and Li 2018 [69]	IgG deficiency	3	NR	No quantitative outcomes Improvements in ASD symptoms and IgG and IgM levels

**Table 7 jpm-11-00488-t007:** Retrospective Uncontrolled Case Studies of Immunoglobulin G Treatment in Autism Spectrum Disorder. Chronic inflammatory Demyelinating Polyneuropathy (CIDP); Common Variable Immunodeficiency (CVID); N-Methyl-D-Aspartic Acid (NMDA); Nicotinic Acetylcholine Receptor (nAChR); Not Reported (NR), month old (mo), years old (yo).

Study	Medical Indication	Participant	IVIG Treatment	Outcomes (Non-Quantitative)
**Immune Abnormality**				
Suez and Scharnwebber 1997 [72]	CVID	15 yo boy	NR	Significant improvement in ASD symptoms
Wang et al., 2005 [73]	CVID	22 yo man	Monthly Dose NR	Significant improvements in appetite, weight gain, and serious infections
Salehi Sadaghiani et al., 2013 [74]	CVID	13 yo boy	NR	No improvements reported
**Inflammatory Neuropathy**				
Sommerville et al., 2007 [75]	CIDP Epilepsy	8 yo boy	0.4 g/kg/d for 5 days	No improvements reported
Kamata et al., 2017 [76]	Inflammatory Neuropathy	6 yo girl	0.4 g/kg/d for 5 days	Neuropathy improved No mention of changes in ASD symptoms
**Immune Mediated Encephalopathy**				
Scott et al., 2014 [77]	anti-NMDA receptor encephalitis	33 mo boy	0.4 g/kg/d for 5 days	Language and social skills improved
Menon et al., 2014 [79]	anti-nAChR receptor encephalitis	5 yo boy	w/plasmapheresis	Improvements in hyperactivity, agitation, speech, and social interaction
Akcakaya et al., 2015, 2016 [70,71]	Enterovirus Encephalitis, Seizures	14 yo girl	0.02 g/kg	Improvements in eye contact, speech, communication, and seizures
Gonzalez-Toro et al., 2013 [78]	anti-NMDA receptor encephalitis	5 yo boy	0.4 g/kg/day for 5 days	Improvements in ASD symptoms and language
Bouboulis and Mast 2016 [80]	Autoimmune Encephalopathy	5 yo boy	1.6 g/kg every 8 weeks for 2 years	Improvements in ASD symptoms, language, learning, and memory
**Other Conditions**				
Xu et al., 2017 [81]	Neuroleptic malignant syndrome	32 yo man	NR	Seizures and dysautonomia improved No mention of changes in ASD symptoms

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
