# Peer review of "A Systematic Review and Meta-Analysis of Immunoglobulin G Abnormalities and the Therapeutic Use of Intravenous Immunoglobulins (IVIG) in Autism Spectrum Disorder"

_jpm, 2021, doi:10.3390/jpm11060488_

Round 1

Reviewer 1 Report

Dear authors, dear editors,

thanks very much for asking me to review the manuscript “A systematic review and meta-analysis of immunoglobulin G abnormalities and the therapeutic use of intravenous immuno-globulins (IVIG) in autism spectrum disorder”, submitted for publication in JPM. The authors have conducted a systematic review and meta-analysis to summarise the evidence for abnormal IgG concentrations and the therapeutic use of IVIG in ASD, also focusing on adverse events.

The study is overall interesting and methodologically strong, considering they followed the most updated PRISMA guidelines. After careful review, I am advising the editors to re-consider the manuscript for publication after minor revisions. I would be keen to read a second draft.

Many regards

A

Detailed comments

-The introduction might benefit from a brief description of the existing literature around the topic of study. For example, could the authors clarify the relationship between lower levels of IgG and more severe aberrant behaviour in children with ASD? Have other systematic reviews on this topic been carried out? What did they find? I think a few paragraphs to ‘set the scene’ would increase the readers’ engagement and motivation to read the full paper.

-The section Methods shall include a paragraph describing how risk of bias was assessed and by whom (1 or 2 reviewers?)

-Could the authors consider adding the total number of participants included in the meta-analyses presented in paragraphs 3.1.2 and 3.2.5

-The discussion/conclusions sections are slightly weak at the moment, since they seem to repeat the results presented in the earlier sections. Could the authors try to summarise the findings in just a paragraph (at the beginning of discussion) and then move on to discuss the implications of such results?

-The conclusion introduces interesting and relevant ideas, which I think should be presented in the introduction. I would probably shorten the conclusion to one sentence (e.g., the sentence starting with “Taken together, IVIG is..”).

Author Response

Comment 1: The introduction might benefit from a brief description of the existing literature around the topic of study. For example, could the authors clarify the relationship between lower levels of IgG and more severe aberrant behaviour in children with ASD? Have other systematic reviews on this topic been carried out? What did they find? I think a few paragraphs to ‘set the scene’ would increase the readers’ engagement and motivation to read the full paper.

Response 1: Many reviews in immunology have been broader, focusing on the widespread immunological abnormalities associated with ASD. We have now expanded the introduction to reference these reviews and discuss potential mechanisms which link immunological abnormalities such as low IgG levels and behavior.

Comment 2: -The section Methods shall include a paragraph describing how risk of bias was assessed and by whom (1 or 2 reviewers?)

Response 2: We have now described in more detail how risk of bias was assessed and specifically stated that it was by both authors. We have also provided detailed assessment of the various biases in the discussion.

Comment 3: -Could the authors consider adding the total number of participants included in the meta-analyses presented in paragraphs 3.1.2 and 3.2.5

Response 3: We apologize if the description of the meta-analysis was not specific enough. We have now specifically stated the total number of participants in the meta-analyses.

Comment 4: -The discussion/conclusions sections are slightly weak at the moment, since they seem to repeat the results presented in the earlier sections. Could the authors try to summarise the findings in just a paragraph (at the beginning of discussion) and then move on to discuss the implications of such results?

Response 4: We have now enhanced the description of our findings and providing some implications of these findings.

Comment 5: -The conclusion introduces interesting and relevant ideas, which I think should be presented in the introduction. I would probably shorten the conclusion to one sentence (e.g., the sentence starting with “Taken together, IVIG is.”).

Response 5: We have shortened the conclusion as suggested and added a sentence to the introduction that foreshadows the findings of the study.

Reviewer 2 Report

The authors sent for publication very interesting article - a systematic review and meta-analysis of immunological disturbances in autistic spectrum disorder (ASD) together with a summary of immunomodulation therapeutical possibilities with intravenous IVIG application.  The article is very detailed and carefully processed. They found, however, heterogeneous results in studies on IgG concentration in ASD and a distinct approach in meta-analysis of immunoglobulin G concentration in patients as well as in controls. Only a few large well-controlled studies were found, and therefore it was difficult to make any firm conclusions. They identified a total of 27 publications in which IVIG were used for treatment, usually due to some ASD immune comorbidities and abnormalities. However, one of the major limitations of these studies were the lack of standardized outcome measures. Most studies did not document the exact dose, frequency, and length of the treatment.

ASD refer to a gamut of developmental disorders impacting communications and social skills, characterized by the expression of restricted repetitive stereotyped behaviors. The spectrum of disorders is extremely wide from the high functioning Asperger syndrome through to pervasive developmental disorder not otherwise specified (PDD-NOS) up to childhood disintegrative disorder with regressive ASD symptoms and deterioration in motor skills. Besides the well-known risk factors such as genetics, prenatal and perinatal factors, neuroanatomical abnormalities and environmental factors, the authors devoted their attention to the immunological domain that surely deserves the attention. Increased number of astrocytes and microglia in the cerebral cortex of ASD patients that provide metabolic and functional support to neurons and act as immune CNS cells, may support the importance of immunological pathophysiological mechanisms. Recently, the question of gut bacteria influence on ASD development has been raised and documented the role of gut-immune-brain axis importance.

The article summarizing contemporary knowledge on immunoglobin G abnormalities and therapeutic use of IVIG in ASD patients is worth publishing and may serve as a background for the future research. If distinct clinical units including in the ASD patients´ group are separated, more uniform results may be found.              

Author Response

Comment 1: ….Most studies did not document the exact dose, frequency, and length of the treatment.

Response 1: We thank the reviewer for the positive comments and have reiterated the limitations outlined by the reviewer for clarity

Comment 2: ….. Increased number of astrocytes and microglia in the cerebral cortex of ASD patients that provide metabolic and functional support to neurons and act as immune CNS cells, may support the importance of immunological pathophysiological mechanisms. Recently, the question of gut bacteria influence on ASD development has been raised and documented the role of gut-immune-brain axis importance.

Response 2: We have added a discussion of the importance of metabolism and the gut microbiome on immunological dysfunction in ASD

Comment 3: The article summarizing contemporary knowledge on immunoglobin G abnormalities and therapeutic use of IVIG in ASD patients is worth publishing and may serve as a background for future research. If distinct clinical units including in the ASD patients´ group are separated, more uniform results may be found.

Response 3: We thank the reviewer for the positive comments and have added comments in the discussion highlighting the importance of well-performed transgenerational studies.

Round 2

Reviewer 1 Report

Dear authors, dear editors,

thanks very much for asking me to review a revised version of the manuscript “A systematic review and meta-analysis of immunoglobulin G abnormalities and the therapeutic use of intravenous immuno-globulins (IVIG) in autism spectrum disorder”, submitted for publication in JPM.

The manuscript was already overall interesting and methodologically strong at submission, and it has been now improved based on our comments. I am advising the editors to accept the manuscript for publication in the journal.

As a note, there are some imprecisions and typos in the manuscript, which should be carefully checked by the authors at proofreading stage.

Many regards

A